# Understanding image motion with group representations

**Andrew Jaegle**∗**, Stephen Phillips**∗**, Daphne Ippolito, and Kostas Daniilidis**
University of Pennsylvania
Philadelphia, PA 19104
{ajaegle, stephi, daphnei, kostas}@seas.upenn.edu

## Abstract

Motion is an important signal for agents in dynamic environments, but learning to represent motion from unlabeled video is a difficult and underconstrained problem. We propose a model of motion based on elementary group properties of transformations and use it to train a representation of image motion. While most methods of estimating motion are based on pixel-level constraints, we use these group properties to constrain the abstract representation of motion itself. We demonstrate that a deep neural network trained using this method captures motion in both synthetic 2D sequences and real-world sequences of vehicle motion, without requiring any labels. Networks trained to respect these constraints implicitly identify the image characteristic of motion in different sequence types. In the context of vehicle motion, this method extracts information useful for localization, tracking, and odometry. Our results demonstrate that this representation is useful for learning motion in the general setting where explicit labels are difficult to obtain.

## 1 Introduction

Motion perception is a key component of biological and computer vision. By understanding how a stream of images reflects the motion of the world around it, an agent can better judge how to act. For example, a fly can use visual motion cues to dodge an approaching hand and to distinguish this threat from a looming landing surface (Reiser & Dickinson (2013)). Motion is an important cue for understanding actions and predicting 3D scene structure, and it has been extensively studied from computational, ethological, and biological perspectives (Hildreth & Koch (1987)).

In computer vision, the problem of motion representation has typically been approached from either a local or global perspective. Local representations of motion are exemplified by optical flow. Flow represents image motion as the 2D displacement of individual pixels of an image, giving rich low-level detail while foregoing a compact representation of the underlying scene motion. In contrast, global representations such as those used in visual odometry attempt to compactly explain the movement of the whole scene. Such representations typically rely on a rigid world assumption, thus limiting their applicability to more general settings.

Image transformations due to motion form a subspace of all continuous image transformations. Smooth changes in the motion subspace correspond to sequences of images with realistic motion. We wish to characterize this subspace. The motion subspace differs from other image transformation subspaces, such as changes in the space of images of human faces. Smooth changes in this space also form a subspace of image transformations, but one containing transformations that do not occur in natural image sequences, such as the face of one person transforming into the face of another. A representation that characterizes motion should be sensitive to the distinction between image transformations that are realistic (produced by image motion) vs. those that are unrealistic (not produced by image motion).

To be useful for understanding and acting on scene motion, a representation should capture the motion of the observer and all relevant scene content. Supervised training of such a representation is challenging: explicit motion labels are difficult to obtain, especially for nonrigid scenes where

---
∗Authors contributed equally.

it can be unclear how the structure and motion of the scene should be decomposed. We propose a framework for learning global, nonrigid motion representations without labels. While most methods of representing motion rely on pixel-level reconstruction or correspondence to guide learning, our method constrains the representation itself by directly addressing the properties of the latent motion space.

Motion has several properties that we use to operationalize to what extent a model characterizes it. (1) A model of motion can be read out to estimate metric properties of the motion in the scene, such as the camera translation and rotation. (2) A model of motion should represent the same motion identically regardless of the image content. For example, the motion of an object moving to the right should be represented the same whether the object is a cat or a dog. (3) A model of motion should distinguish sequences produced by natural motion from sequences with image transitions not produced by natural motion, such as cuts.

Here, we present a general model of visual motion and describe how the group properties of visual motion can be used to constrain learning in this model (Figure 1). We enforce the group properties of associativity and invertibility during training using a metric learning approach (Chopra et al. (2005)) on recomposed sequences. We describe how this technique can be used to train a deep neural network to represent the motion in image sequences of arbitrary length in a low-dimensional, global fashion. We present evidence that the learned representation captures the global structure of motion in both 2D and 3D settings without labels, hard-coded assumptions about the scene, or explicit feature matching.

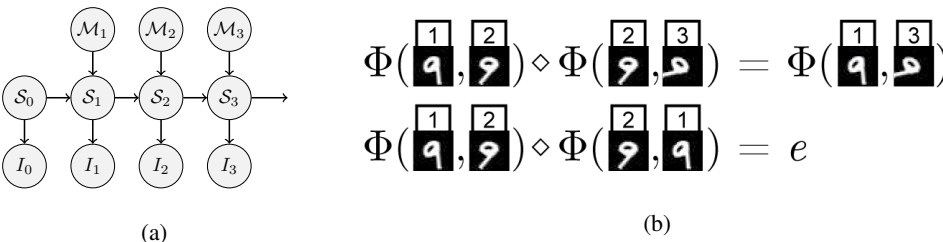

(a)  (b)

Figure 1: **(a)** A graphical model describing the relationship between the latent scene structure $\{\mathcal{S}_t\}$, motion $\{\mathcal{M}_t\}$, and the observed images of a sequence. We describe a method for learning a representation $\overline{\mathcal{M}}$ of the motion space $\mathcal{M}$ from observed image sequences $\{I_t\}$. **(b)** By recomposing sequences of images to satisfy the group properties of associativity and invertibility, we construct pairs of image sequences with equivalent motion. We use these properties to learn an approximate group homomorphism $\Phi \in \overline{\mathcal{M}}$ between motion in the world and in an embedding.

## 2 RELATED WORK

### 2.1 MOTION REPRESENTATIONS

The most common global representations of motion are structure from motion (SfM) and simultaneous localization and mapping (SLAM), which represent motion as a sequence of poses in $\mathcal{SE}(3)$ perhaps along with a static point cloud (Scaramuzza & Fraundorfer (2011), Fraundorfer & Scaramuzza (2012)). These approaches have achieved great success in many applications in recent years, but they are unable to represent non-rigid or independent motions. The most commonly used local representation is optical flow, which estimates pixel-wise motion over the image, typically constraining it with a smoothness prior (Sun et al. (2010)). Scene flow (Wedel et al. (2008)) and non-rigid structure from motion (Xiao et al. (2004)) represent a larger class of 3D motions by generalizing optical flow to the estimation of 3D point trajectories. These methods represent motion only at local regions (typically single points) and do not attempt to compactly capture the overall motion.

More similar to our approach is work designing or learning spatio-temporal features (STFs) (Laptev (2005)). STFs are localized and flexible enough to represent non-rigid motions. They typically include a dimensionality reduction step and hence are somewhat global in purview. Recent work has used convolutional neural nets (CNNs) to learn task-related STFs directly from images, including Tran et al. (2015) and Le (2013). Unlike our work, both of these approaches are restricted to fixed

temporal windows of representation. Taylor et al. (2010) uses a standard unsupervised learning technique to learn spatiotemporal features useful for action recognition but not for motion itself.

## 2.2 LEARNING REPRESENTATIONS USING VISUAL STRUCTURE

Several recent works have used knowledge of the geometric or spatial structure of images or scenes to train representations. Doersch et al. (2015) trains a CNN to classify the correct configuration of image patches to learn the relationship between an image's patches and its semantic content. The resulting representation can be fine-tuned for image classification. Yu et al. (2016) and Patraucean et al. (2016) train networks to estimate optical flow using the brightness constancy assumption and a smoothness prior as a learning signal. Zhu et al. (2017) and Ren et al. (2017) learn flow using a similar technique. As with other flow based methods, these works use photometric, local constraints. Garg et al. (2016) uses the relationship between depth and disparity to learn to estimate depth from a rectified stereo camera pair with a known baseline. Similarly, Konda & Memisevic (2014) treats motion as a latent variable and exploits the relationship between motion and depth to estimate depth.

Other works that learn from sequences typically focus on static image content rather than motion. Of these, the most similar to ours is Misra et al. (2016), which shuffles the order of images in a sequence to learn representations of image content. Their approach is designed to capture single image properties that are correlated with temporal order rather than motion itself and their shuffling procedure does not preserve the group properties forming the basis of our learning technique. A related approach is slow feature analysis (Wiskott & Sejnowski (2002)), which is motivated by the notion that slowly varying latents are often behaviorally relevant. Other works exploring learning from sequences include Jayaraman & Grauman (2015), which learns a representation equivariant to the egomotion of the camera, and Agrawal et al. (2015), which learns to represent images by explicitly regressing the egomotion between two frames. Instead of learning to represent motion, these works use labeled motion as a learning cue.

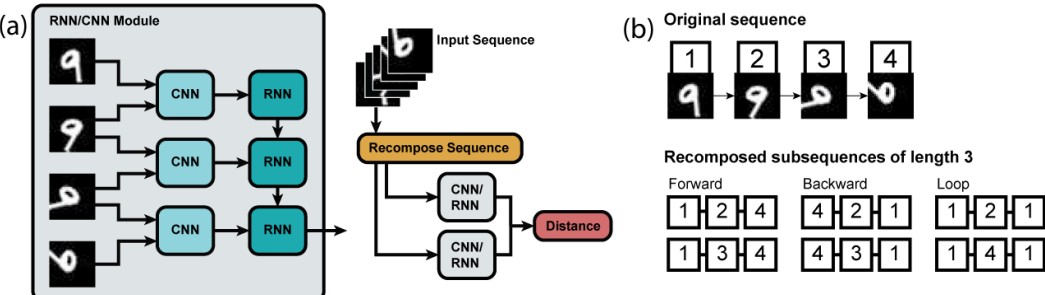

Figure 2: **(a)** Network structure. The RNN output at the final step of the sequence is treated as the sequence embedding. During training, the distance between sequence embeddings is adjusted using an embedding loss. **(b)** We recompose sequences to enforce associativity and invertibility. Sequences with equivalent motion (e.g. 1-2-4 and 1-3-4) serve as positive examples, while sequences with inequivalent motions (e.g. 1-2-4 and 4-3-1) serve as negative examples.

## 3 APPROACH

### 3.1 GROUP PROPERTIES OF MOTION

We base our method on the observation that the set of 3D scene motions, equipped with the composition operation, forms a group. This group describes the latent structure of transformations in continuous, real-world image sequences. By learning an embedding that captures the transformations in scenes that occur during motion, we approximate a group homomorphism between the latent motion of the scene and a representation of this motion. We design our method to capture associativity and invertibility, which allows us to reason about how motions relate and can be composed.

To see that a latent motion space respects these properties, first consider a latent structure space $\mathcal{S}$. In this model, an element of the structure space generates images $\mathcal{I}$ via a projection operator

$\pi : \mathcal{S} \to \mathcal{I}$. We also define a latent motion space $\mathcal{M}$, which is some closed subgroup of the set of homeomorphisms on $\mathcal{S}$. For any element $S$ of the structure space $\mathcal{S}$, a continuous motion sequence $\{M_t \in \mathcal{M} \mid t \geq 0\}$ generates a continuous image sequence $\{I_t \in \mathcal{I} \mid t \geq 0\}$ where $I_t = \pi(M_t(S))$. For a discrete set of images, we can rewrite this as $I_t = \pi((M_{\Delta_t} \circ M_{t-1})(S)) = \pi(M_{\Delta_t}(S_{t-1}))$, which defines a hidden Markov model, as illustrated in Figure 1 (a). As $\mathcal{M}$ is a closed subgroup of the homeomorphisms on $\mathcal{S}$, it is associative, it contains the identity, and all of its elements have unique inverses in the group.

A simple, specific case of this model is rigid image motion, such as the motion produced by a camera translating and rotating through a rigid scene in 3D. Here, the latent structure of the scene $\mathcal{S}$ can be modeled by a point cloud with a motion space given by $\mathcal{M} = \mathcal{SE}(3)$. For a scene with $N$ rigid bodies, we can describe the motion with a tuple of $\mathcal{SE}(3)$ values, $\mathcal{M} = (\mathcal{SE}(3))^N$, where the $N$th motion acts on the set of points belonging to the $N$th rigid body. Generalizing $\mathcal{M}$ to general homeomorphisms gives the most general case of motion. As different scenes contain different degrees of freedom and affordances, it is typically unclear which group effectively characterizes motion in a given real-world setting. We propose to learn this in a data-driven manner.

## 3.2 Learning motion by group properties

Our goal is to learn a function $\Phi : \mathcal{I} \times \mathcal{I} \to \overline{\mathcal{M}}$ that maps pairs of images to a representation $\overline{\mathcal{M}}$ of the motion space $\mathcal{M}$. We also learn a corresponding composition operator $\diamond : \overline{\mathcal{M}} \to \overline{\mathcal{M}}$ that emulates the composition of elements in $\mathcal{M}$. This representation and operator should respect the properties of the motion group in question.

We exploit the structure of the domain to learn to represent and compose motions without labels. If an image sequence $\{I_t\}$ is sampled from a continuous motion sequence, then the sequence representation should have the following properties for all times $t_0, t_1, t_2, t_3$, where $t_0 < t_1 < t_2 < t_3$, reflecting the group properties of the latent motion:

(i) Associativity: $\Phi(I_{t_0}, I_{t_2}) \diamond \Phi(I_{t_2}, I_{t_3}) = (\Phi(I_{t_0}, I_{t_1}) \diamond \Phi(I_{t_1}, I_{t_2})) \diamond \Phi(I_{t_2}, I_{t_3}) = \Phi(I_{t_0}, I_{t_1}) \diamond (\Phi(I_{t_1}, I_{t_2}) \diamond \Phi(I_{t_2}, I_{t_3})) = \Phi(I_{t_0}, I_{t_1}) \diamond \Phi(I_{t_1}, I_{t_3})$. The motion of differently composed subsequences of a sequence are equivalent.

(ii) Existence of the identity element: $\Phi(I_{t_0}, I_{t_1}) \diamond e = \Phi(I_{t_0}, I_{t_1}) = e \diamond \Phi(I_{t_0}, I_{t_1})$, and $e = \Phi(I_t, I_t)$ for any $t$. Null image motion corresponds to the (unique) identity in the latent space.

(iii) Invertibility: $\Phi(I_{t_0}, I_{t_1}) \diamond \Phi(I_{t_1}, I_{t_0}) = e$. The motion of a reversed image sequence is the inverse of the motion of the original image sequence.

We use an embedding loss to approximately enforce associativity and invertibility among subsequences sampled from an image sequence. Associativity is encouraged by pushing differently composed sequences with equivalent motion to the same representation. Invertibility of the representation is encouraged by pushing each forward sequence away from its backward counterpart and by pushing all loops to the same representation (i.e. to a learned representation of the identity in the embedding space). We encourage the uniqueness of the identity representation by pushing loops away from non-identity sequences in the representation. Because loops have equivalent (identity) motion regardless of scene content, we also push together loops drawn from different sequences. This procedure is illustrated schematically in Figure 2.

Learning in this framework can be viewed as inference on the graphical model in Figure 1 (a). Learning a representation of motion is an underconstrained problem, and the group learning rules we introduce here constrain the problem with minimal restriction on the types of scene changes that can be embedded.

In contrast, in optical flow, inference is constrained using the brightness constancy assumption, which assumes that the illumination of a projected scene point does not change between frames (Horn & Schunck (1981)). Our framework encompasses flow inference if brightness constancy is viewed as a constraint on the projection operator $\pi$. However, the brightness constancy assumption is invalid in many settings. Our model's assumptions about geometric properties of motion in the world are valid even over large motions and changing illumination.

The latent structure and motion of a scene are in general non-identifiable, which implies that for any given scene, there are several $\overline{\mathcal{M}}$ that can adequately represent $\mathcal{M}$. We do not claim to learn a unique representation of motion, but rather we attempt to capture one such representation. Our method assumes the scene has a relatively stable structure, and we do not expect it to handle rapidly changing content or sequence cuts. We also expect our method to have difficulty representing motion in cases of temporally extended occlusion due to the unobservability of motion in such settings.

### 3.3 SEQUENCE LEARNING WITH NEURAL NETWORKS

The functions $\Phi$ and $\diamond$ are implemented as a convolutional neural network (CNN) and a recurrent neural network (RNN), respectively. We use a long short-term memory (LSTM) RNN (Hochreiter & Schmidhuber (1997)) due to its ability to reliably learn over long time sequences. The input to the network is in an image sequence $[I_1, ..., I_t]$. The CNN $\Phi$ processes these images and outputs an intermediate representation $[C_{1,2}, ..., C_{t-1,t}]$. The LSTM operates over the sequence of CNN outputs to produce an embedding sequence $[R_{1,2}, ..., R_{t-1,t}]$. We treat $R(\{I_t\}) = R_{t-1,t}$ as the embedding of sequence $\{I_t\}$. This configuration is illustrated schematically in Figure 2 (a).

Table 1: Average embedding error (equation 1) on held-out data. Results are averaged over forward, backward, and loop sequences. Errors are relative to a chance error of 1: values lower than 1 indicate that equivalent (inequivalent) motions are close together (far apart) in the embedding space.

| CNN input method | Motion condition | MNIST | KITTI |
|---|---|---|---|
| Image pairs | Equivalent | 8.1e−4 | 7.2e−3 |
| Image pairs | Inequivalent | 1.7e−2 | 8.0e−2 |
| Single image | Equivalent | 0.74 | 3.5e−2 |
| Single image | Inequivalent | 0.79 | 3.5 |

The network is trained to minimize a hinge loss with respect to the embedding of pairs of sequences:

$$L(R^1, R^2) = \begin{cases} d(R^1, R^2), & \text{if positive pair} \\ \max(0, m - d(R^1, R^2)), & \text{if negative pair} \end{cases} \quad (1)$$

where $d(R^1, R^2)$ measures the distance between the embeddings of two example sequences $\{I_t^1\}$ and $\{I_{t'}^2\}$, and $m$ is a fixed scalar margin. Positive examples are image sequences that are compositionally equivalent, while negative examples are those that are not. We use the cosine distance for all experiments (with $m = 0.5$), as it is smooth and discourages learning the trivial embedding. In early experiments, results with an L2 distance were similar.

We include six recomposed subsequences for each training sequence: two forward, two backward, and two identity subsequences, as shown in Figure 2 (b). Subsequences are sampled such that all three sequence types share some of their frames. To discourage the network from paying attention to only the beginning or end of a sequence, we use several image recomposing schemes. Forward and backward sequences are sampled to either have the same or different starting frames, and they are drawn from either the same subsequence or from temporally adjacent subsequences. Because the network is exposed to sequences with the same start and end frames but different motion, this sampling procedure encourages the network to rely on features in the motion domain, rather than on static differences. During training, we use sequences of varying length to encourage generalization to motions of different temporal scale.

We also explored learning a representation $\Phi$ taking single images (and not image pairs) as CNN input. Because the CNN in this configuration only has access to single images, it cannot extract image motion directly. In all domains we tested, the representation learned from image pairs outperformed the one learned from single images (Table 1).

## 4 EXPERIMENTS

We first demonstrate that our learning procedure can discover the structure of motion in the context of rigid scenes undergoing 2D translations and rotations. We then show that our method learns features useful for representing motion on KITTI (Geiger et al. (2012)), a dataset of vehicle sequences with motion due to the camera and independent objects. In all experiments, networks were trained using

Adam (Kingma & Ba (2014)). For MNIST training, we used a fixed decay schedule of 30 epochs and with a starting learning rate chosen by random search (1e-2 was a typical value). For MNIST, typical batch sizes were 50-60 sequences, and for KITTI (Geiger et al. (2012)) the batch sizes were typically 25-30 sequences. All networks were implemented in Torch (Collobert et al. (2011)).

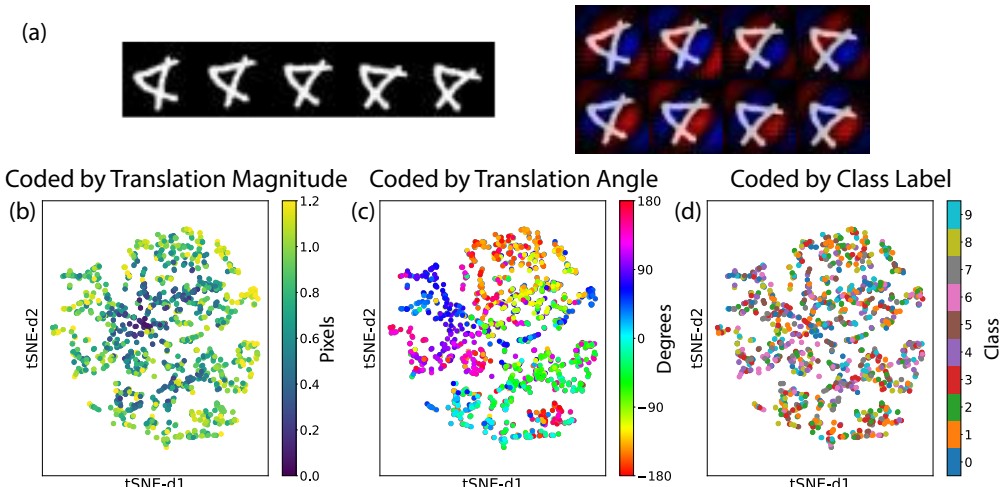

Figure 3: **(a)** An example test sequence from MNIST and the corresponding saliency maps. Saliencies show the gradient backpropagated from the final RNN timestep. Each column represents an image pair passed to one of the CNNs. **(b)-(d)** tSNE of the network embedding on the test set, with points labeled by **(b)** the magnitude of translation in pixels, **(c)** the translation direction in degrees, and **(d)** the digit label (0-9). The representation clusters sequences by both translation magnitude and direction but not identity.

We use dilated convolutions to obtain large receptive fields suitable for capturing large-scale motion patterns. We used ReLU nonlinearities and batch normalization (Ioffe & Szegedy (2015)) after each convolutional layer. CNN output was passed to an LSTM with 256 hidden units, followed by a linear layer with 256 hidden units. In all experiments, CNN-LSTMs were trained on sequences 3-5 images in length. We tested MNIST networks with sequences of up to 12 images with similar results.

## 4.1 RIGID MOTION IN 2D

To test the ability of our learning procedure to represent motion, we trained a network on a dataset consisting of image sequences created from the MNIST dataset. Each sequence consists of images undergoing a smooth motion drawn from the group of 2D translations and rotations ($\mathcal{SE}(2)$) for 20 frames. We sampled transformation parameters uniformly from $[-10, 10]$ pixels for both horizontal and vertical translation and from $[0, 360)$ degrees for rotation. Validation errors are given in Table 1.

We visualize the representation learned on this data using tSNE (van der Maaten & Hinton (2008)) on the sequence embedding for test images undergoing a random translation (Figure 3). The network representation clearly clusters sequences by both the direction and magnitude of translation. No obvious clusters appear in terms of the image content. Similar results were obtained when test data included both translation and rotation. This suggests that the network has learned a representation that captures the properties of motion in the dataset. This content-invariant clustering occurs even though the network was never trained to compare images with different spatial content and the same motion.

To further probe the network, we visualize image-conditioned saliency maps in Figure 3. These saliency maps show the positive (red) and negative (blue) gradients of the network activation with respect to the input image. As discussed in Simonyan et al. (2013), such a saliency map can be interpreted as a first-order Taylor expansion of the function $\Phi$, evaluated at image $I$. The saliency map thus gives an indication of how pixel values affect the representation. These saliency maps show gradients with respect to the output of the LSTM over the sequence.

Table 2: Linear regression from the learned embedding to the translation and rotation of the KITTI odometry dataset consistently performs better than chance (guessing the mean value). Table entries show mean squared error $\pm$ standard error (percent improvement).

|  | Translation X (cm) | Translation Y (cm) | Translation Z (cm) |
|---|---|---|---|
| Mean | 5.92 $\pm$ 1.5e-01 | 3.01 $\pm$ 1.2e-01 | 1904.75 $\pm$ 3.1e+01 |
| Ours | 5.05 $\pm$ 1.2e-01 (14.71%) | 2.83 $\pm$ 1.2e-01 (6.10%) | 1539.04 $\pm$ 2.3e+01 (19.20%) |
| Flow+PCA (4 PCs) | 3.18 $\pm$ 9.5e-02 (46.27%) | 2.92 $\pm$ 1.2e-01 (3.47%) | 1754.36 $\pm$ 2.8e+01 (7.91%) |
| Flow+PCA (256 PCs) | 1.89 $\pm$ 8.5e-02 (68.07%) | 2.32 $\pm$ 1.2e-01 (23.42%) | 239.46 $\pm$ 5.6e+00 (87.43%) |

|  | Rotation X (deg) | Rotation Y (deg) | Rotation Z (deg) |
|---|---|---|---|
| Mean | 0.02 $\pm$ 3.3e-04 | 0.98 $\pm$ 1.6e-02 | 0.03 $\pm$ 4.8e-04 |
| Ours | 0.02 $\pm$ 3.1e-04 (4.03%) | 0.79 $\pm$ 1.5e-02 (19.12%) | 0.02 $\pm$ 4.0e-04 (21.28%) |
| Flow+PCA (4 PCs) | 0.02 $\pm$ 3.3e-04 (1.16%) | 0.29 $\pm$ 3.3e-03 (70.11%) | 0.03 $\pm$ 4.8e-04 (0.17%) |
| Flow+PCA (256 PCs) | 0.00 $\pm$ 9.8e-05 (79.33%) | 0.05 $\pm$ 1.7e-03 (94.50%) | 0.01 $\pm$ 1.3e-04 (82.53%) |

Intriguingly, these saliency maps bear a strong resemblance to the spatiotemporal energy filters of classical motion processing (Adelson & Bergen (1985)), which are known to be optimal for 2D speed estimation in natural scenes under certain assumptions Burge & Geisler (2015). We note that these saliency maps do not simply depict the shape of the filters of the first layers, but rather represent the implicit filter instantiated by the full network on this image. When compared across different frames, it becomes clear that the functional mapping learned by the network can flexibly adapt in orientation and arrangement to the image content, unlike standard energy model filters.

Table 3: Interpolation distances on KITTI (as in Figure 4), averaged across test data. Distances are consistently lower for the true frame than for visually similar frames (inside sequence) and dissimilar frames (outside sequence) when using the embedding, but not the Euclidean distance.

| Method | Skipped frames | True middle frame | Inside (min value) | Outside (min value) |
|---|---|---|---|---|
| Embedding | 1 | 3.91e-03 | 7.67e-02 | 2.94e-01 |
| Euclidean | 1 | 7.92e-04 | 7.97e-04 | 1.09e-03 |
| Embedding | 2 | 1.18e-02 | 2.02e-02 | 1.34e-01 |
| Euclidean | 2 | 9.59e-04 | 8.13e-04 | 1.08e-03 |

## 4.2 REAL-WORLD MOTION IN 3D

We use the KITTI dataset (Geiger et al. (2012)) to test the model's representation of motion in 3D scenes with camera and independent motion. We use the representation trained on KITTI tracking in all experiments. First, we evaluate how well it can decode camera motion. We compute the representation on all two-frame sequences of KITTI visual odometry, which are labeled with ground truth camera poses. We then linearly regress from these representations to the change in camera pose between the frames using least squares.

For comparison, we show results using a recent self-supervised flow algorithm Yu et al. (2016). The output of this method is a dense optical flow field. In order to regress from this flow field to camera poses, we downsample the flow fields and run principal component analysis (PCA) over the full training set. We then linearly regress from the flow field PCA components to the camera motion parameters using least squares. Flow fields are computed at a resolution of 320x96 pixels, and PCA is computed on downsampled flow fields of size 160x48 pixels. We include up to 256 PCA components in the regression. We refer to this method as Flow+PCA.

Results on held-out test data are displayed in Table 2. Despite not being trained on any ground truth pose and not seeing any data from the odometry dataset, the learned representation decodes pose consistently better than chance (guessing the mean value). The largest improvements are in X and Z translation, which also exhibit the most variance in the KITTI odometry dataset. These results are not competitive with the Flow+PCA results or state-of-the-art odometry methods, but they suggest our method recovers a meaningful representation of motion. On KITTI visual odometry, our method

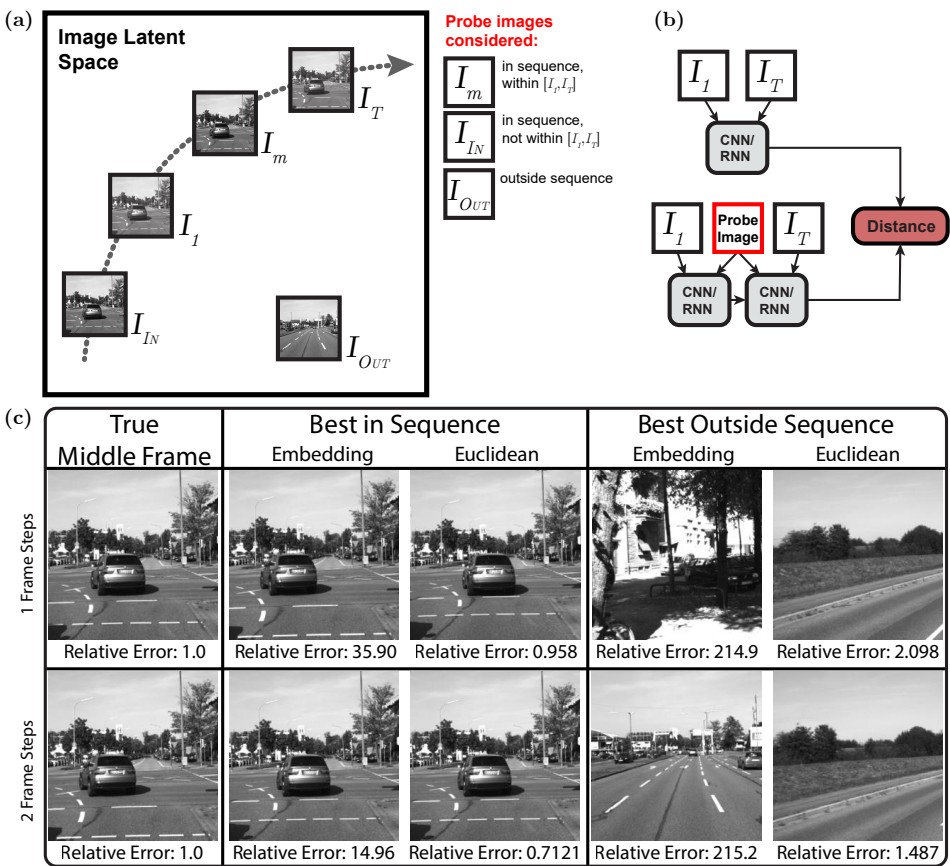

Figure 4: **(a)** Natural image motion is a subspace of the space of all image transformations, and a particular motion can be viewed as a path in the latent space of natural images. Although $[I_1, I_K, I_T]$ has a total transformation equivalent to $[I_1, I_T]$ for any value of $K$, only $[I_1, I_m, I_T]$ can be produced by natural image motion. **(b)** In interpolation experiments, we compare the distance between the embedding of $[I_1, I_T]$ with the embedding of this sequence after inserting either a true middle frame ($I_m$) or another frame ($I_{\text{IN}}$ or $I_{\text{OUT}}$). **(c)** Images with lowest relative error taken from the sequence or from the whole dataset, for each distance measure. Errors are relative to that of the true middle frame in the corresponding measure: high relative errors ($\gg 1$) indicate the distance distinguishes realistic motion from unrealistic motion. Images other than true middle frame produce dramatically higher errors when using the embedding but not when using a Euclidean distance.

performs similarly to regression to the Flow+PCA method using four to five principal components 6, which suggests that it is able to capture the dominant global components of motion.

Next, we test the ability of our network to capture the typical motion of the scene by quantifying its performance on an interpolation task. Given an image sequence $[I_1, ..., I_T]$, we compute the distance between the embedding of the first and last frames, $R([I_1, I_T])$, vs. the sequence composed of the first frame, a middle frame, and the last frame $R([I_1, I_m, I_T])$ (Figure 4). By comparing the distance when using the true middle frame with the distance when using a different middle frame, we can estimate how sensitive the network is to deviations from the typical dynamics of natural scenes, and hence how well it has learned the relevant motion subgroup. Results for the full KITTI tracking dataset are shown in Table 3. We compare our method to a Euclidean distance, computed as the mean pixelwise distance between the probe image and either $I_1$ or $I_T$ (whichever is smaller). Note that the embedding distance of the true frame is dramatically lower than that of all other frames. This does not hold for the Euclidean distance, which is often lower for non-interpolating images in the sequence, and is not dramatically different for frames from other sequences.

Finally, we visualize saliency maps on an example sequence in the KITTI dataset in Figure 5. The saliency map highlights objects moving in the background and the independent motions of the car

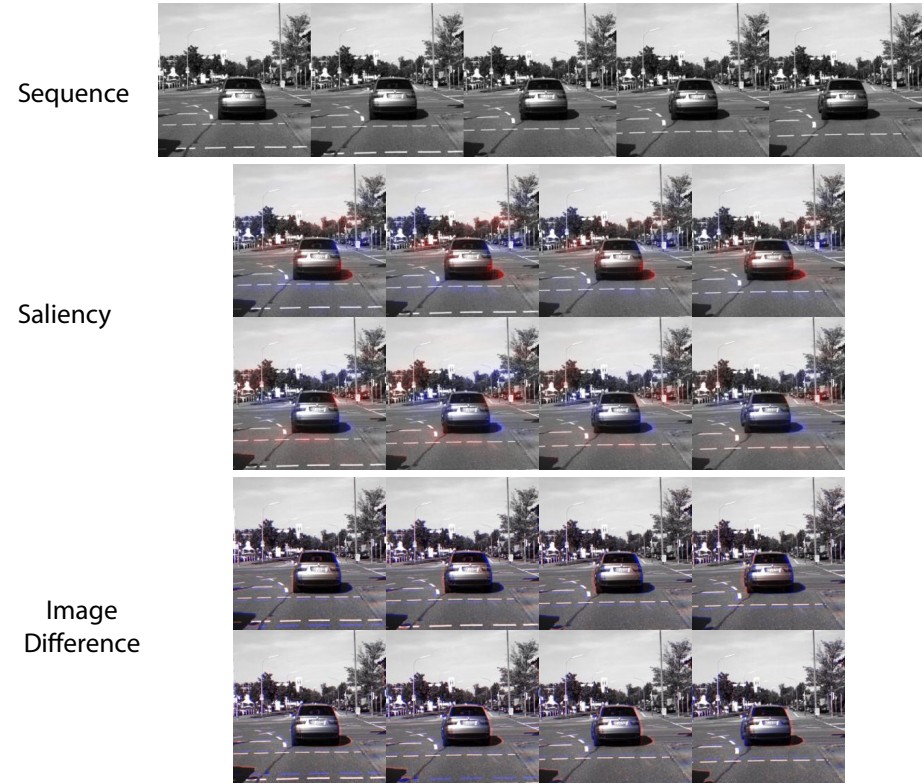

Figure 5: Saliency results on a test sequence from KITTI tracking with both camera and independent motion. The network focuses on areas that are relevant to determining motion in 3D, not simply regions with large temporal image gradients.

in the foreground. The network highlights areas of the car that can move, such as the bumper, even when these areas don't contain prominent image differences. These results suggest our method may be useful for learning features for independent motion detection and tracking.

There are few standard tasks for directly evaluating motion methods beyond odometry. We attempted to regress from our learned representation to action classes but were unable to obtain competitive performance. This is not surprising: previous work has shown spatial features are more discriminative for this task, and motion features require extensive processing to be useful (e.g. Simonyan & Zissermann (2014)). In future work, we will explore using group properties to encourage intermediate latents to represent motion along with other tasks. We expect that an embedding that maintains representations of spatial content alongside representations of motion will be more successful in settings like action recognition that depend on both sets of features.

## 5  CONCLUSION

We have presented a new model of motion and a method for learning motion representations. We have shown that enforcing group properties of motion is sufficient to learn a representation of image motion. These results suggest that this representation is able to generalize between scenes with disparate content and motion to learn a motion state representation useful for navigation, prediction, and other behavioral tasks relying on motion. Because of the wide availability of unlabeled video sequences in many settings, we expect our framework to be useful for generating better global motion representations in a variety of real-world tasks.

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

## SUPPLEMENTAL: ADDITIONAL EXPERIMENTS

Here, we expand on the comparison to the self-supervised optical flow baseline given in table 2. Our method performs equivalently to the Flow+PCA method using the top four to five principal components (which account for most of the motion variance on KITTI, as shown in Figure 7). The marginal improvement in Flow+PCA appears to sharply drop off beginning around four to five principal components as well. As we saw before, the most dramatic increases in performance come from the Z component of translation and the Y component of rotation, which are the axes of the dominant motion and where chance error is highest.

We note that egomotion estimation benefits greatly from maintaining information about spatial position. Methods using flow fields maintain the information by explicitly representing local motion at each position of the image, but our method is global and does not. KITTI visual odometry is characterized by stereotyped depth and is reasonably modeled as rigid. Under these circumstances, camera translation and rotation can be estimated from a full flow field nearly linearly (Heeger & Jepson (1992)). Consistent with this explanation, flow principal components appear to capture both the dominant motions exhibited by the vehicle on this dataset and the stereotyped depth configuration of KITTI (Figure 8). The good performance of Flow+PCA here highlights the clear advantage of domain-restricted models and learning rules in a setting where those domain restrictions are appropriate. Our learning rule and model do not make these more restrictive assumptions but still performs reasonably in this setting.

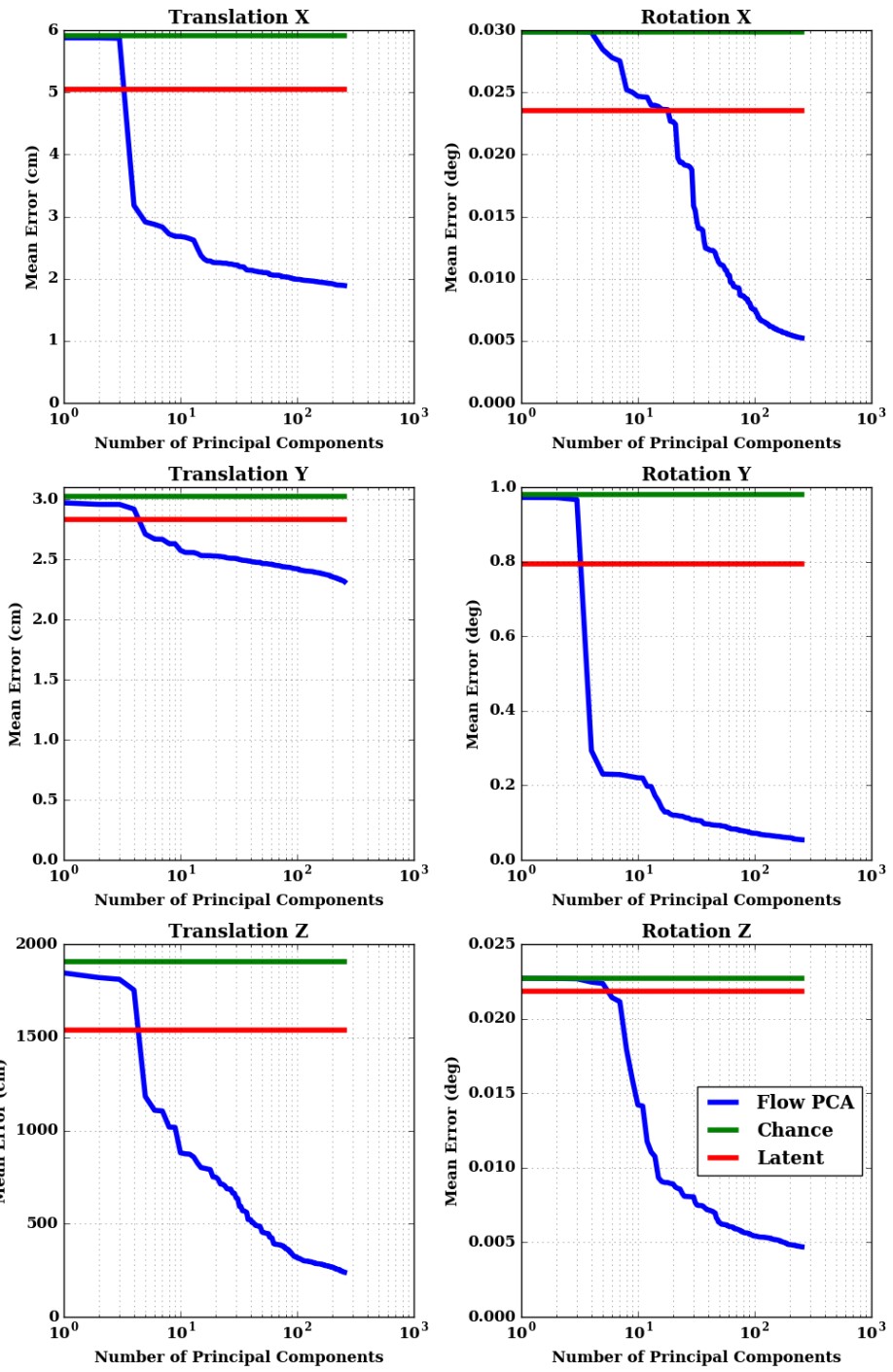

Figure 6: Error on egomotion regression from self-supervised flow PCA as a function of the number of principal components included. Horizontal lines reflect our method (latent, shown in red) and a chance baseline (show in green).

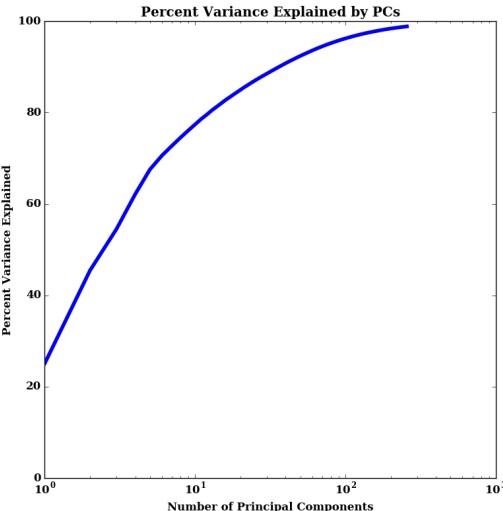

Figure 7: Cumulative percent variance explained of the optical flow in KITTI odometry as a function of the number of principal components included. 67% of the variance is explained by the first 5 components; 90% of the variance is explained by the first 40 principal components.

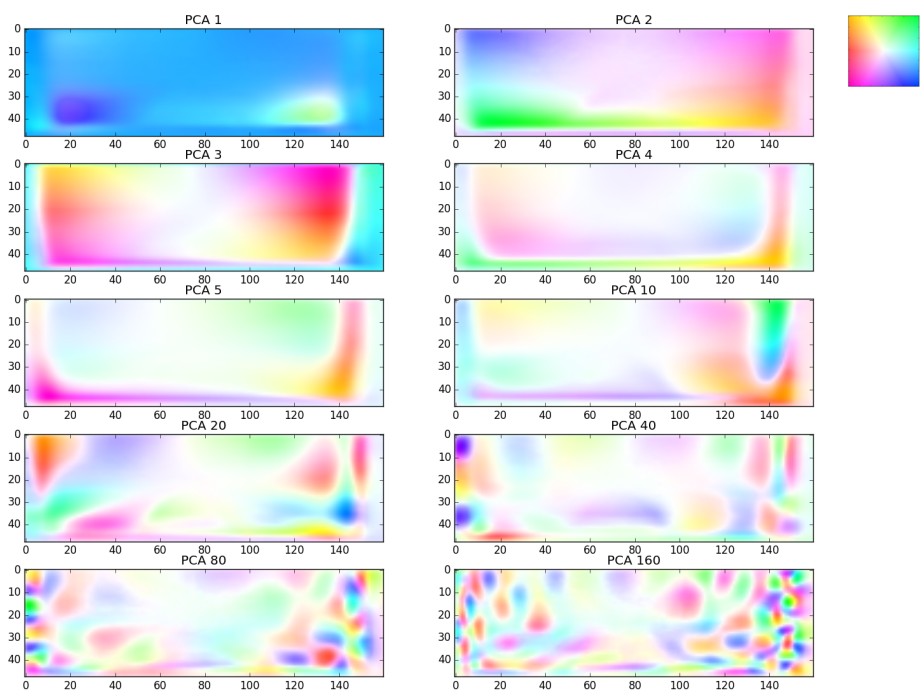

Figure 8: Representative principal components of optical flow on the KITTI odometry dataset. The first few components capture the dominant motions (forward and left/right turning) and reflect the stereotypical depth structure of KITTI.

