# OpenReview forum: "Understanding image motion with group representations "
_ICLR.cc/2018/Conference — Accept (Poster)_

### Official Review · AnonReviewer1 · 2017-11-23
**Interesting approach to estimate motion**

**Rating:** 7
**Confidence:** 3

**Review:**

The authors propose to learn the rigid motion group (translation and rotation) from a latent representation of image sequences without the need for explicit labels.
Within their data driven approach they pose minimal assumptions on the model, requiring the group properties (associativity, invertibility, identity) to be fulfilled.
Their model comprises CNN elements to generate a latent representation in motion space and LSTM elements to compose these representations through time.
They experimentally demonstrate their method on sequences of MINST digits and the KITTI dataset.

Pros:
- interesting concept of combining algebraic structure with a data driven method
- clear idea development and well written
- transparent model with enough information for re-implementation
- honest pointers to scenarios where the method might not work well

Cons:
- the method is only intrinsically evaluated (Tables 2 and 3), but not compared with results from other motion estimation methods

---

> ### Author Response · Authors · 2018-01-05
> **Response to AnonReviewer1**
>
> Thank you for your review and comments. We have added a comparison to a self-supervised optical flow method to better contextualize our method. See the response to AnonReviewer3 for more details (under the heading Compare to unsupervised optical flow).

---

### Official Review · AnonReviewer3 · 2017-11-27
**Interesting idea, but more detailed experimental validation is needed.**

**Rating:** 5
**Confidence:** 4

**Review:**

Paper proposes an approach for learning video motion features in an unsupervised manner. A number of constraints are used to optimize the neural network that consists of CNN + RNN (LSTM). Constraints stem from group structure of sequences and include associativity and inevitability. For example, forward-backward motions should cancel each other out and motions should be additive. Optimized network is illustrated to produce features that can be used to regress odometry.

Overall the approach is interesting from the conceptual point of view, however, experimental validation is very preliminary. This makes it difficult to asses the significance and viability of the approach. In particular, the lack of direct comparison, makes it difficult to asses whether the proposed group constraints are competitive with brightness constancy (or similar) constraints used to learn motion in an unsupervised manner in other papers.

It is true that proposed model may be able to learn less local motion information, but it is not clear if this is what happens in practice. In order to put the findings in perspective authors should compare to unsupervised optical flow approach (e.g., unsupervised optical flow produced by one of the proposed CNN networks and used to predict odometer on KITTI for fair comparison). Without a comparison of this form the paper is incomplete and the findings are difficult to put in the context of state-of-the-art.

Also, saying that learned features can predict odometry “better than chance” (Section 4.2 and Table 2) seems like a pretty low bar for a generic feature representation.

---

> ### Author Response · Authors · 2018-01-05
> **Response to AnonReviewer3**
>
> Thank you for your helpful comments and suggestions. We have added a comparison to a recent self-supervised optical flow approach for the KITTI visual odometry experiments as suggested and updated the text accordingly. See below for responses to specific comments.
>
> -Compare to unsupervised optical flow-
> To put our method in context, we have included comparisons to a recent method for self-supervised optical flow estimation (Yu et al 2016). The output of this method is a dense optical flow field. In order to regress from this flow field to the camera motion parameters, we downsample the flow fields and run PCA over the full training set of fields. We then linearly regress from the flow field PCs to the camera motion parameters using least squares. Flow fields are computed at a resolution of 320x96, and PCA is computed on downsampled flow fields of resolution 160x48. The results from this method on KITTI are now included in the paper in table 2 and figure 6 (in the supplement).
>
> The full optical flow method outperforms our method. We note that egomotion estimation benefits greatly from maintaining information about spatial position (which a flow field does, but our method does not). KITTI is characterized by stereotyped depth and is reasonably modelled as rigid, and under these circumstances camera translation and rotation can be estimated from a full flow field nearly linearly. The good performance of self-supervised flow + PCA here highlights the clear advantage of domain-restricted models and learning rules in a setting where those domain restrictions are appropriate. Our learning rule and model do not make these more restrictive assumptions but still performs reasonably in this setting.
>
> To further contextualize our method, we also show the flow results as a function of the number of flow PCs included. As shown in figure 6 (in the supplement), our method outperforms the flow method up to four flow PCs, and outperforms estimates of x-dimension rotation up to around ten flow PCs. These results bolster our claim of learning a reasonable representation of motion with minimal domain assumptions.
>
> -Not clear it learns less local motion representation-
> Thank you for drawing attention to this point. Unlike standard models of motion, our model is designed so that it can capture nonlocal, nonrigid motion in principle. By contrast, standard models are designed to capture only local motion (optical flow) or global motion with rigid structure (egomotion). In practice, it appears that our learning rule does not succeed at capturing all of the nonlocal structure that the model can support. However, the KITTI results show that our trained model can be used to linearly regress reasonable estimates of camera translation and rotation, suggesting that the representation is nonlocal to some extent. The model and learning rule proposed here should serve as a baseline for future, more powerful learning rules that are still able to model nonrigid, nonlocal motion.
>
> -"Better than chance" as a low bar.-
> Our results show the feasibility and limits of what can be learned about motion using a model that makes as few assumptions as possible about image motion and by using a minimal learning rule to impose these constraints. As we mention, our method isn't competitive with state of the art results. To improve the interpretability of this method, we have included a systematic comparison of our method with a self-supervised optical flow method as a function of the amount of flow information included in the regression (table 2 and figure 6 (supplemental)).

---

### Official Review · AnonReviewer2 · 2017-11-28
**The proposed unsupervised losses appear weak**

**Rating:** 4
**Confidence:** 4

**Review:**

The paper presents a method that given a sequence of frames, estimates a corresponding motion embedding to be the hidden state of an RNN (over convolutional features) at the last frame of the sequence. The parameters of the motion embedding are trained to preserve properties of associativity  and invertibility of motion, where the frame sequences have been recomposed (from video frames) in various way to create pairs of frame sequences with those -automatically obtained- properties. This means, the motion embedding is essentially trained without any human annotations.
Experimentally, the paper shows that in synthetic moving MNIST frame sequences motion embedding discovers different patterns of motion, while it ignores image appearance (i.e., the digit label). The paper also shows that linear regressor trained in KITTI on top of the unsupervised motion embedding to estimate camera motion performs better than chance.
Q to the authors: what labelled data were used to train the linear regressor in the KITTI experiment?
Empirically, it appears that supervision by preserving group transformations may not  be immensely valuable for learning motion representations.



Pros
1)The neural architecture for motion embedding computation appears reasonable
2)The paper tackles an interesting problem

Cons
1)For a big part of the introduction the paper refers to the problem of `` ````"learning motion” or `''understanding motion”  without being specific what it means by that.
2)The empirical results are not convincing of the strength of imposing group transformations for self-supervised learning of motion embeddings.
3)The KITTI experiment is not well explained as it is not clear how this regressor was trained to predict egomotion out of the motion embedding.

---

> ### Author Response · Authors · 2018-01-05
> **Response to AnonReviewer2**
>
> Thank you for your helpful comments. We have modified the paper to better explain the points you mention, and we hope this clarifies the text. See below for responses to specific comments.
>
> -What labelled data for linear regressor, how was KITTI trained? experiment not well explained-
> For the regression experiments, we use the sequences of the KITTI visual odometry benchmark, which include annotations of camera translation and rotation. To test our model on this dataset, we linearly regress from the learned representation to the camera motion between an image pair. We do not fine-tune our model with labelled camera motion, but perform linear regression using least squares on the learned representations. The representations themselves are trained only on KITTI tracking, a different subset of KITTI than we use for least squares. Results are shown on the full set of KITTI visual odometry sequences. We also use least squares to regress from optical flow PCA components in the updated experiments (we discuss this in more depth in the response to reviewer 3, under the heading Compare to unsupervised optical flow).
>
> We have also expanded figure 4 to explain more clearly the design of the KITTI interpolation experiments. The purpose of this experiment is to test whether our method is sensitive to deviations from the motion subspace - i.e. whether is sensitive to the difference between realistic and subtly unrealistic image sequences.
>
> -"learning motion", "understanding motion"-
> Thank you for pointing out the ambiguity in these terms. By understanding motion, we mean learning a model that characterizes image transformations due to structure-preserving changes in time. Not all changes in image sequences reflect motion (e.g. other changes are due to cuts in a video, etc.), and here we are concerned with characterizing the subspace of image transformations due specifically to motion. We have added a clarification of these terms in the updated manuscript.
>
> Motion has several properties that we use to operationalize to what extent a learned representation characterizes the motion subspace. (1) A representation that characterizes motion can be read out to estimate the motion in the scene. In particular, it can be used to regress metric properties of motion, such as camera translation and rotation. We test this prediction by regressing to camera translation and rotation on KITTI and by tSNE clustering on MNIST digits, which reveals clustering by translation. (2) The model should also represent the same motion identically regardless of the image content. For example, a motion of a digit moving at one pixel per frame to the right should be represented the same whether the digit is a "5" or a "3". We demonstrate this with the tSNE clustering results in the paper. (3) Image sequences produced by natural motion should be represented differently than image sequences not produced by natural motion. That is, a representation that characterizes motion can distinguish realistic motion from unrealistic motion. We demonstrate this property in our interpolation experiments.
>
> -Strength of group representations-
> We agree that stronger, domain-specific learning rules can be obtained by incorporating more constraints for a specific context. The smoothness constraint used in optical flow is one such rule in the context of locally rigid motion. Here we show that the more general rules based on groups can lead to representations useful for motion.
>
> The group-based learning rules are complementary to more domain-specific learning rules, but they are also applicable in settings where domain-specific rules like smoothness or brightness constancy are not appropriate. We hope that our results will be useful to future work designing learning rules incorporating both more generic and more domain-specific inductive biases for learning motion and other kinds of image transformations.

---

### Author Response · Authors · 2018-01-05
**Changes made to the paper**

- Added a paragraph to the introduction expanding explanation of motion and its relationship to image transformations.
- Added a paragraph to the introduction explaining how we operationalize the goal of understanding motion with a model.
- Added a paragraph to section 4.2 of experiments section describing the Flow+PCA method and interpreting results.

- Expanded Table 2 with the results on the Flow+PCA baseline.
- Revised Table 3 caption for clarity.
- Modified figure 4 image and caption to better explain the experiment and interpretation.

- Added a supplemental section with more extensive description of Flow+PCA experiments along with interpretation and comparison to the group-based method. Three new figures added (Figures 6, 7, and 8).

---

### Decision · Program_Chairs · 2018-01-29
**ICLR 2018 Conference Acceptance Decision**

**Decision:**

Accept (Poster)

**Comment:**

An interesting model, for an interesting problem but perhaps of limited applicability - doesn't achieve state of the art results on practical tasks.
Paper has other limitations, though the authors have addressed some in rebuttals.